# An ecological study to evaluate the association of Bacillus Calmette-Guerin (BCG) vaccination on cases of SARS-CoV2 infection and mortality from COVID-19

Lucy Chimoyi[1]⊙*, Kavindhran Velen[1,2]⊙, Gavin J. Churchyard[1,3], Robert Wallis[1], James J. Lewis[4]⊙, Salome Charalambous[1,3]⊙

1 The Aurum Institute, Parktown, Johannesburg, South Africa, 2 Sydney School of Medicine (Central Clinical School), Faculty of Medicine and Health, University of Sydney, Sydney, Australia, 3 School of Public Health, University of the Witwatersrand, Parktown, Johannesburg, South Africa, 4 Y Lab, Public Services Innovation Lab for Wales, School of Social Sciences, Cardiff University, Wales, United Kingdom

⊙ These authors contributed equally to this work.
* lchimoyi@auruminstitute.org

**Data Availability Statement:** All relevant data are within the manuscript and its Supporting information files.

## Abstract

As the SARS-CoV2 pandemic has progressed, there have been marked geographical differences in the pace and extent of its spread. We evaluated the association of BCG vaccination on morbidity and mortality of SARS-CoV2, adjusted for country-specific responses to the epidemic, demographics and health. SARS-CoV2 cases and deaths as reported by 31 May 2020 in the World Health Organization situation reports were used. Countries with at least 28 days following the first 100 cases, and available information on BCG were included. We used log-linear regression models to explore associations of cases and deaths with the BCG vaccination policy in each country, adjusted for population size, gross domestic product, proportion aged over 65 years, stringency level measures, testing levels, smoking proportion, and the time difference from date of reporting the 100th case to 31 May 2020. We further looked at the association that might have been found if the analyses were done at earlier time points. The study included 97 countries with 73 having a policy of current BCG vaccination, 13 having previously had BCG vaccination, and 11 having never had BCG vaccination. In a log-linear regression model there was no effect of country-level BCG status on SARS-CoV2 cases or deaths. Univariable log-linear regression models showed a trend towards a weakening of the association over time. We found no statistical evidence for an association between BCG vaccination policy and either SARS-CoV2 morbidity or mortality. We urge countries to rather consider alternative tools with evidence supporting their effectiveness for controlling SARS-CoV2 morbidity and mortality.

## Background

The global spread of the novel coronavirus (SARS-CoV2) has been unprecedented, with 188 countries now reporting at least one case by the 25 June 2020. The total number of confirmed

**Funding:** The author(s) received no specific funding for this work.

**Competing interests:** The authors have declared that no competing interests exist.

cases globally has increased rapidly, with 9.3 million people diagnosed with SARS-CoV2 as reported by the World Health Organization (WHO) [1]. The initial wave of SARS-CoV2 was linked primarily to travel from affected countries, however subsequent increases are the result of sustained community transmission [2]. As the pandemic has progressed, there have been marked geographical differences in the pace and extent of its spread. In the early stages of the pandemic, exponential growth was predominantly seen in high income countries, while low- and middle-income countries had not yet seen the same exponential increases in new cases, often demonstrating longer periods for reaching cumulative milestones e.g. time taken to reach 100 confirmed cases. Countries like Iran and Netherlands reached 100 confirmed cases of SARS-CoV2 within 10 days of the first case, however Cambodia took almost 60 days to reach the same milestone [3].

Considerable underlying variation in drivers of the epidemic may explain observed differences in morbidity among low- and middle-income countries compared to high-income countries, such as age distribution, health systems strength and ultimately the varied response to the epidemic by respective countries including testing rates and stringency of lockdown measures. One specific theory has been the population-level impact of Bacillus Calmette-Guerin (BCG) vaccination, which is thought by some to have conferred a degree of protection to SARS-CoV2 in countries where it was or is currently part of the routine immunization schedule. The BCG vaccine is administered to newborns and offers protection against disseminated forms of TB in children, including TB meningitis and miliary TB [4]. It contains a live attenuated strain of *Mycobacterium bovis*, to prevent tuberculosis disease, but it is also used as immune stimulant in other conditions [5].

Since the introduction of BCG, epidemiological studies have demonstrated that the vaccine reduced infant mortality independent of its effect on tuberculosis [4, 6, 7]. These beneficial effects have been called 'heterologous' or 'non-specific' effects [4] and may offer an explanation for significant variations in SARS-CoV2 morbidity and mortality in low- and middle-income countries, many of whom still continue to use BCG. However, the effect of BCG on TB is only effective in children and this has warranted trials to determine whether revaccination of adolescents could reduce the incidence of TB infection as the effect clearly wanes in adolescents [8]. Association for protection in elderly is further questioned since intravesicular BCG is an effective therapy for bladder cancer [9], but infant BCG vaccination has not been associated with protection against bladder cancer in the elderly.

A few unpublished ecological studies have suggested that there may be an association between BCG and the SARS-CoV2 epidemic [10, 11], while others have discredited its link to mortality and morbidity [12]. Countries with and without BCG vaccination policy tend to differ markedly in terms of their economies, health systems and exposure to infectious diseases [13]. In addition, these studies all assessed BCG association very early in the pandemic and likely underestimated confounding, particularly differences in the timing of the epidemic and temporal responses at a country-level [9]. Other factors that require evaluation include whether BCG vaccination was recently discontinued, so that most people over 30 years of age have been vaccinated, the testing rates for COVID affecting case incidence and the age structure of the population [14].

Overcoming the SARS-CoV2 pandemic will require continued understanding of the disease and potential innovation on multiple fronts. Disparities in available resources globally necessitate that while we strive to find new interventions, we should consider existing tools. Understanding the role of BCG in limiting morbidity and mortality may offer insights into its renewed use and vaccination status may serve as a factor for risk stratification in the ongoing response. The aim of our study is to evaluate the association of historical or ongoing BCG vaccination on morbidity and mortality of SARS-CoV2, embedded within country-specific

responses to the epidemic, demographics and health policies. In addition, to understand differing results observed in various published and unpublished papers, we will evaluate trends in association as epidemic data accumulated.

## Methods

### Country selection

The analysis for both cases and deaths was conducted based on reporting of cumulative cases and deaths by 31 May 2020. In order to give sufficient follow-up time since SARS-CoV-2 became "established" in each country, countries were included if they had reported at least of SARS-CoV-2 or at least 10 SARS-CoV-2-related deaths by 3 May 2020, giving at least 28 days of follow up by 31 May 2020. In addition, we excluded countries with no existing information on BCG as described below and/or a population size <1 million as number of actual cases would be low in such countries.

### Data extraction

Table 1 indicates sources for each indicator that were explored. The data on SARS-CoV2 was extracted from the daily WHO Situation Reports. SARS-CoV2 cases and deaths as reported by 31 May 2020 on the WHO situation reports were used [15]. SARS-CoV2 testing numbers were obtained on the 11 June 2020 from the Worldometer website [16].

Information on BCG vaccination policies were obtained from the World BCG Atlas which is an online BCG database that is hosted by McGill University [13, 17]. Where these data were not available on the BCG atlas, we searched immunization schedules of countries for information, and if BCG was not currently universally used (for five countries), we assumed that it had never been part of their schedule [18]. These were confirmed by checking the UNICEF estimates of immunisation for these countries [19]. In cases, where the standardized immunization schedule could not be found, the countries were removed from the analysis (n = 10). Immunization coverage in most countries that do have universal access is above 90% and so that was not considered as a variable for analysis.

For SARS-CoV2 stringency levels, we used the Oxford COVID-19 Government Response Tracker [20], which classifies country-level stringency levels according to containment and closure policies, such as school closures and restrictions in movement and determines a stringency index from 1 to 100 on a daily basis for each country. We used the stringency index level for each country at the date of reporting the 100th confirmed case and then at a date 28 days post this milestone. Smoking data were obtained from the World Health Organisation [21]. TB incidence was obtained from the 2019 TB Global Report [22]. The raw data on which our analyses are based is available in Supplementary Tables.

### Data analysis

BCG status was categorized to current BCG policy, previous BCG policy (where it was in place for greater than 30 years) and no BCG policy (where it was never used or used for less than 30 years) [13]. The overall proportion smoking was determined by using the proportions of males and females in the population and the proportion of smokers in each gender group. We calculated the country-specific time difference from the date of the 100th reported case to 31st May 2020 to control for the stage of the epidemic that each country was in.

We log-transformed the continuous outcome variables of country-level cases and deaths, using log-linear regression to examine the relationship between the outcome (COVID cases and mortality) and BCG status in univariable models. In the earliest stages of an epidemic the

**Table 1. Indicators included in article and their sources.**

| | Indicator | Source of information | Organisation |
|---|---|---|---|
| | Name of Country | WHO situation report: https://www.who.int/emergencies/diseases/novel-coronavirus-2019/situation-reports | World Health Organisation |
| **COVID-19 indicators** | Date of 1st case | WHO situation report | World Health Organisation |
| | Date of 10th death | WHO situation report | World Health Organisation |
| | 28 days after 10th death | WHO situation report | World Health Organisation |
| | Testing numbers (on 29th April) | Worldometers https://www.worldometers.info/coronavirus/#countries | |
| | Number of people infected by 28 days post 10th death | WHO situation report | World Health Organisation |
| | Number of people dead by 28 days post 10th death | WHO situation report | World Health Organisation |
| **BCG indicators** | Availability of a current BCG policy | BCG atlas: http://www.bcgatlas.org/index.php | Mc Gill International TB Centre |
| | Date/year of first BCG vaccine administration after policy implementation | BCG atlas | Mc Gill International TB Centre |
| | Date/year of last BCG vaccine administration after policy implementation | BCG atlas | Mc Gill International TB Centre |
| | Coverage of BCG | BCG atlas | Mc Gill International TB Centre |
| | Strain of the Bacilli in the vaccine | BCG atlas | Mc Gill International TB Centre |
| | Age at administration | BCG atlas | Mc Gill International TB Centre |
| **Other country level indicators** | Restrictions—stringency level at 10th death | https://www.bsg.ox.ac.uk/research/research-projects/coronavirus-government-response-tracker | University of Oxford/ Blatvatnik School of Government |
| | Restrictions—stringency level at 28 days after 10th death | https://www.bsg.ox.ac.uk/research/research-projects/coronavirus-government-response-tracker | University of Oxford/ Blatvatnik School of Government |
| | World Bank income level (High, Medium, Low) | https://www.worldbank.org/en/country | World Bank |
| | Gross Domestic Product | https://www.worldbank.org/en/country | World Bank |
| | Net immigration | https://www.worldbank.org/en/country | World Bank |
| | Air transport, registered carrier departures worldwide | https://data.worldbank.org/indicator/IS.AIR.DPRT?view=map | World Bank |
| | Population of country | https://www.worldbank.org/en/country | World Bank |
| | Population density of country | https://www.worldbank.org/en/country | World Bank |
| | Proportion of population over 65 years | https://www.worldbank.org/en/country | World Bank |
| | TB incidence rate | World TB report 2019 | World Health Organisation |
| | TB mortality rate | World TB report 2019 | World Health Organisation |
| | Medical doctors per capita | https://www.worldbank.org/en/country | World Bank |
| | Nurses per capita | https://www.worldbank.org/en/country | World Bank |
| | % smoking among men | https://apps.who.int/gho/data/node.main.65 | World Health Organisation |
| | % smoking among women | https://apps.who.int/gho/data/node.main.65 | World Health Organisation |

population size of a country is not a determinant of initial spread. However, as the epidemic progresses, the population size can become a strong determinant of numbers of cases and deaths. Hence, we used the numbers of cases and numbers of deaths as outcomes, rather than the incidence rate and mortality rate. We adjusted for the population size in regression models by using the log of the population size, and so a regression coefficient for log population size of zero would correspond to no association between cases or deaths and population size, while a regression coefficient of 1 would be equivalent to using incidence and mortality rates as the outcomes.

We constructed adjusted models by including variables that confounded the associations between outcomes and BCG status. We built these final models adding one by one each

variable starting with the one that demonstrated the strongest confounding of the associations. Other data considered, but not included in the model based on a lack of observed association with SARS-CoV2 cases or mortality, were TB incidence rates per country for 2019, HIV prevalence, medical doctors per capita, nurses per capita, population density, proportion of urban population, net immigration, number of air passengers per year and stringency index at 28 days.

Finally, we repeated the log linear regression models for each of SARS-CoV2 cases and deaths using the numbers of SARS-CoV2 cases and deaths reported at three prior time-points during the epidemic (mid-April, end of April, and. mid-May 2020) to better understand comparisons with results from previously unpublished studies. Analyses were performed in Stata version 14 [23] and maps were drawn using ArcMap version 10.7.1 [24]. Unadjusted regression coefficients are presented, with a regression coefficient of zero indicating no association.

## Results

There were 107 countries for which complete data on the total SARS-CoV2 cases were available end of May 2020. Data sources are summarised in Table 1. Ten countries were excluded for having a population <1 million or no data on BCG vaccination, leaving 97 countries in the analysis (Fig 1). Country-level characteristics stratified by BCG status are shown in S1–S3 Tables.

Table 2 shows the log-linear regression models for cases and deaths at 31 May 2020 adjusted for population, gross domestic product, age under 65, stringency measures, testing levels, smoking proportion, and the time difference from date of 100[th] reported case to 31 May. Upon adjustment, there was no statistical evidence for an association between number of cases and BCG status (Table 2). However, positive associations were observed with population size (coefficient per log increase: 0.93; 95% CI: 0.54, 1.13, p-value<0.0001) and tests per capita (coefficient per log increase 0.72; 95% CI: 0.45, 0.99, p-value<0.0001). An inverse relationship between SARS-CoV2 cases and strength of stringency measures at 100 cases was observed (coefficient: -0.02; 95% CI: -0.03, -0.003, p-value = 0.01). The level of stringency measures implemented at the 100th case for each country are visualised in Fig 2 showing countries in

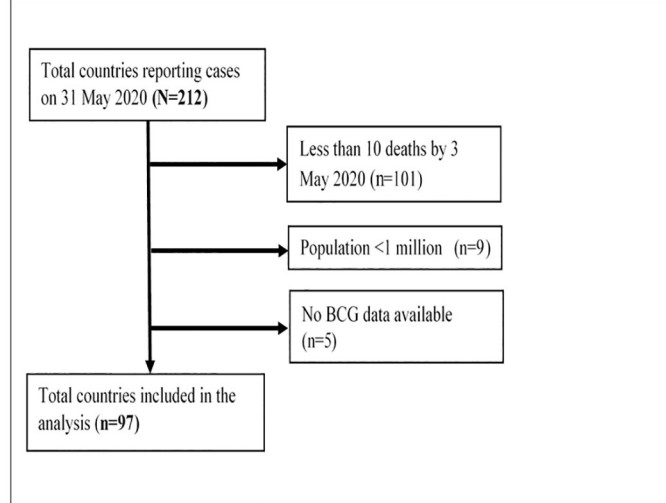

**Fig 1. Flow chart describing country selection.**

**Table 2. Country-level adjusted associations between BCG status and log-transformed number of SARS-CoV2 cases and deaths at 31 May 2020.**

| | SARS-CoV2 cases (n = 85) | | | SARS-CoV2 deaths (n = 84) | | |
|---|---|---|---|---|---|---|
| | Estimate | 95% CI | p-value | Estimate | 95% CI | p-value |
| BCG vaccination status | | | | | | |
| Current | Ref | - | - | Ref | - | - |
| Ever | -0.17 | -1.00, 0.65 | 0.68 | 0.35 | -0.74, 1.44 | 0.52 |
| Never | -0.48 | -1.28, 0.33 | 0.24 | 0.06 | -1.01, 1.12 | 0.92 |
| **Controlled for:** | | | | | | |
| Log gross domestic product per capita ($)[1] | -0.02 | -0.41, 0.37 | 0.93 | -0.10 | -0.62, 0.42 | 0.70 |
| Log population size (millions) [2] | 0.93 | 0.54, 1.31 | <0.0001 | 1.20 | 0.69, 1.72 | <0.0001 |
| Proportion of population over 65 years (%)[3] | -0.03 | -0.08, 0.02 | 0.20 | 0.08 | 0.01, 0.14 | 0.02 |
| Log tests per capita[4] | 0.72 | 0.45, 0.99 | <0.0001 | 0.44 | 0.08, 0.80 | 0.02 |
| Stringency level at 100[th] case | -0.02 | -0.03, -0.003 | 0.01 | -0.01 | -0.03, 0.002 | 0.08 |
| Smoking prevalence[5] | -0.01 | -0.03, 0.01 | 0.36 | -0.01 | -0.04, 0.02 | 0.48 |
| Difference between date of 100[th] case and end of May 2020 (days) | 0.04 | -0.01, 0.08 | 0.10 | -0.04 | 0.10, 0.02 | 0.21 |

[1]Estimates as at 2018;

[2]Estimates as at 2018;

[3]Estimates as at 2018;

[4]Estimates as at 11 June 2020;

[5]Estimates as at 2018

red having a low level of stringency measures (>25%) and in green, a higher level of stringency measures (>75%) implemented.

Similarly, for deaths, there was no statistical evidence for an association between number of deaths and BCG status after adjusting for selected variables (Table 2). However, positive associations were observed with population size (coefficient per log increase: 1.20, 0.69, 1.72, p-value<0.0001), proportion of population aged over 65 years (coefficient per percentage point increase 0.08, 95%CI 0.01, 0.14; p-value = 0.02), and tests per capita (coefficient per log increase 0.44, 95% CI 0.08, 0.80, p-value = 0.02). Weak statistical evidence was seen for an

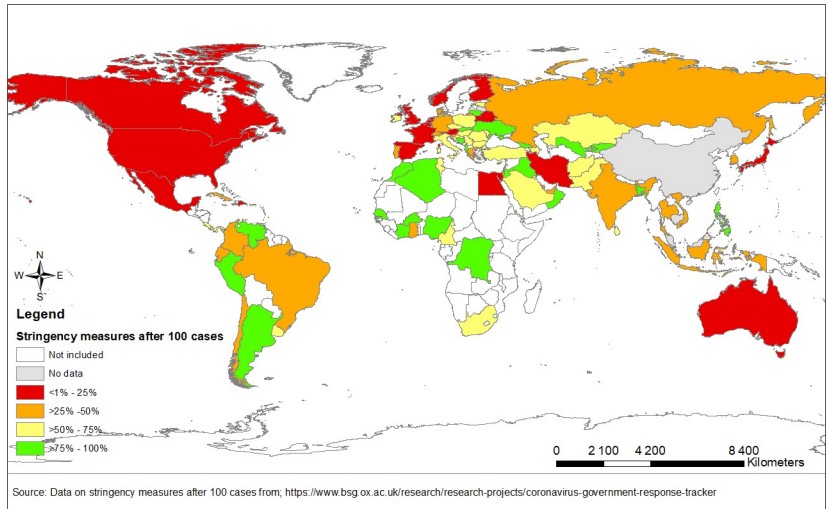

**Fig 2. Stringency measures applied to selected countries after 100 reported COVID-19 cases.**

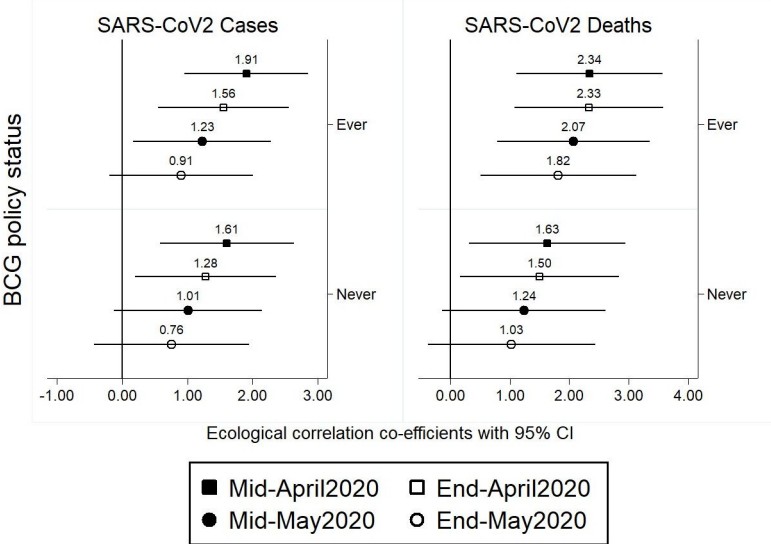

**Fig 3. Univariable associations between cases and deaths with BCG policy at four time points from mid-April to end of May 2020.**

inverse relationship between SARS-CoV2 deaths and strength of stringency measures at 100 cases (coefficient: -0.01, 95% CI: -0.03, 0.002, p-value = 0.08).

To understand why previous studies reported an association, we show the univariable associations between each of cases and deaths with BCG policy at four time points from mid-April to end of May 2020 (Fig 3). We show a trend for a weakening effect on confirmed SARS-CoV2 cases and deaths from mid-April to end of May 2020. These are further explored in multivariable models for cases and deaths at different time points in Tables 3 and 4, with no statistical evidence seen for association between either cases or deaths and BCG policy at any of the time points in adjusted models.

## Discussion

Using data from 97 countries, we evaluated the effect of BCG vaccination policy on SARS-CoV2 morbidity and mortality and found no statistical evidence for an association, with strengths of associations weakening over time as the epidemic progressed. In addition, when evaluating associations with morbidity and mortality at multiple time points over a two-month period, we observed little variation in magnitude and direction of association for all variables except BCG vaccination policy, suggesting that multiple country-level factors may likely underpin the SARS-CoV2 epidemic rather than BCG vaccination policy.

The presence of confounders such as older age, gross domestic product, differential testing and population size were investigated. Similar to other studies which showed higher mortality in older age groups >65 years, our study reported this association which confirms age as a major risk factor for COVID-19 morbidity and mortality at a country level [10, 12]. Although high income countries have robust healthcare systems, the income level of a country was not identified as a confounding factor that explained the association between BCG vaccination and COVID-19 risk. The level of healthcare preparedness including the number of tests conducted was likely to determine the morbidity and mortality cases.

In our ecological analysis, we found that BCG vaccination policy did not influence the magnitude of SARS-CoV2 morbidity and mortality at a country-level. This was confirmed by

Table 3. Country-level adjusted associations between BCG status and log-transformed number of SARS-CoV2 cases.

| | Mid-April 2020 (n = 85) | | | End of April 2020 (n = 85) | | | Mid-May 2020 (n = 85) | | |
|---|---|---|---|---|---|---|---|---|---|
| | Estimate | 95% CI | p-value | Estimate | 95% CI | p-value | Estimate | 95% CI | p-value |
| BCG vaccination status | | | | | | | | | |
| Current | Ref | - | - | Ref | - | - | Ref | - | - |
| Ever | 0.21 | -0.42, 0.85 | 0.51 | -0.02 | -0.72, 0.69 | 0.96 | -0.10 | -0.88, 0.68 | 0.79 |
| Never | 0.25 | -0.36, 0.87 | 0.42 | -0.11 | -0.80, 0.58 | 0.75 | -0.30 | -1.07, 0.46 | 0.43 |
| **Controlled for**: | | | | | | | | | |
| Log gross domestic product per capita ($)[1] | -0.17 | -0.48, 0.14 | 0.28 | -0.09 | -0.43, 0.24 | 0.58 | -0.08 | -0.45, 0.29 | 0.67 |
| Log population size (millions) | 0.85 | 0.56, 1.15 | <0.0001 | 0.87 | 0.53, 1.20 | <0.0001 | 0.92 | 0.56, 1.29 | <0.0001 |
| Proportion of population over 65 years (%) | 0.04 | -0.001, 0.07 | 0.06 | 0.02 | -0.02, 0.06 | 0.37 | -0.01 | -0.05, 0.04 | 0.79 |
| Log tests per capita | 0.65 | 0.44, 0.85 | <0.0001 | 0.68 | 0.45, 0.91 | <0.0001 | 0.71 | 0.44, 0.96 | <0.0001 |
| Stringency level at 100[th] case | -0.01 | -0.02, 0.001 | 0.10 | -0.01 | -0.02, -0.001 | 0.03 | -0.01 | -0.03, -0.003 | 0.01 |
| Smoking prevalence | -0.01 | -0.02, 0.01 | 0.57 | 0.01 | -0.03, 0.01 | 0.42 | -0.01 | -0.03, 0.01 | 0.38 |
| Difference between date of 100[th] case and end of May 2020 (days) | 0.07 | 0.04, 0.10 | <0.0001 | 0.06 | 0.02, 0.09 | <0.001 | 0.05 | 0.005, 0.09 | 0.03 |

[1]Estimates as at 2018;

[2]Estimates as at 2018;

[3]Estimates as at 2018;

[4]Estimates as at 11 June 2020;

[5]Estimates as at 2018

observational studies in Sweden [25], Israel [26] and Germany [27] which used individual level data to test this association. In Sweden, BCG vaccination did not reduce cases or hospitalizations due to COVID-19 in cohorts whose recorded births were before and after 1975 [25]. In Israel, there was no association after comparison between rates of coronavirus PCR test

Table 4. Country-level adjusted associations between BCG status and SARS-CoV2 deaths.

| | Mid-April 2020 (n = 84) | | | End of April 2020 (n = 84) | | | Mid-May 2020 (n = 84) | | |
|---|---|---|---|---|---|---|---|---|---|
| | Estimate | 95% CI | p-value | Estimate | 95% CI | p-value | Estimate | 95% CI | p-value |
| BCG vaccination status | | | | | | | | | |
| Current | Ref | - | - | Ref | - | - | Ref | - | - |
| Ever | 0.55 | -0.48, 1.59 | 0.29 | 0.62 | -0.38, 1.61 | 0.22 | 0.50 | -0.55, 1.55 | 0.34 |
| Never | 0.50 | -0.51, 1.51 | 0.33 | 0.47 | -0.50, 1.45 | 0.33 | 0.27 | -0.75, 1.30 | 0.60 |
| **Controlled for**: | | | | | | | | | |
| Log gross domestic product per capita ($)[1] | -0.17 | -0.66, 0.32 | 0.50 | -0.17 | -0.65, 0.30 | 0.47 | -0.15 | -0.65, 0.35 | 0.55 |
| Log population size (millions) | 1.01 | 0.53,1.50 | <0.0001 | 1.16 | 0.69, 1.62 | <0.0001 | 1.21 | 0.71, 1.69 | <0.0001 |
| Proportion of population over 65 years (%) | 0.08 | 0.02, 0.14 | 0.01 | 0.10 | 0.05, 0.16 | 0.001 | 0.10 | 0.04, 0.16 | 0.002 |
| Log tests per capita | 0.44 | 0.09, 0.78 | 0.01 | 0.43 | 0.10, 0.76 | 0.01 | 0.43 | 0.08, 0.77 | 0.02 |
| Stringency level at 100[th] case | -0.01 | -0.02, 0.01 | 0.24 | -0.01 | -0.03, 0.004 | 0.13 | -0.01 | -0.03, 0.003 | 0.10 |
| Smoking prevalence | -0.001 | -0.03, 0.03 | 0.96 | -0.01 | -0.03, 0.02 | 0.64 | -0.01 | -0.04, 0.02 | 0.51 |
| Difference between date of 100[th] case and end of May 2020 (days) | -0.08 | -0.13, -0.02 | 0.01 | -0.05 | -0.11, 0.0004 | 0.05 | -0.05 | -0.10, 0.01 | 0.13 |

[1]Estimates as at 2018;

[2]Estimates as at 2018;

[3]Estimates as at 2018;

[4]Estimates as at 11 June 2020;

[5]Estimates as at 2018

positivity among Israelis with symptoms suspicious for COVID-19 who did and did not receive BCG vaccination as part of routine childhood immunization in the early 1980s [26]. Lastly, Lindestam Arlehamn et al showed no correlation between BCG coverage and the mortality while comparing COVID-19 deaths Eastern and Western Germany states [27]. A recent systematic review and meta-analysis that evaluated BCG vaccination on SARS-CoV2 infection among 13 full-text articles (12 were not peer-reviewed), reported a similar conclusion despite significant heterogeneity in study-specific findings [28]. Our findings demonstrated that timing of the analyses influenced BCG association, a factor which may explain why earlier studies showed some association, however as more data became available, its significance resolved. The magnitude of the SARS-CoV2 epidemic has and will likely be determined by country-level responses. Initial and ongoing suppression strategies, effective testing and tracing approaches coupled with strong healthcare systems, will ultimately flatten the epidemic curve. The use of BCG vaccination may not be associated with protection against SARS-CoV2, however, the importance of its use as a key TB prevention strategy should not be neglected. We thus support recent calls by the TB community for responsible stewardship of BCG and its continued use in populations against TB, where substantial proven benefits exist [29].

Notwithstanding the limited biological inference of BCG protection against SARS-CoV2, many countries that have succumbed considerably in terms of morbidity and mortality have a previous BCG vaccination policy or are currently vaccinating infants. Our study found that almost 90% of countries we evaluated implemented historic or current BCG vaccination policy. In addition, countries classified as previous or no BCG vaccination policy are all in high-income settings, with the exception of Lebanon. If we assume BCG was capable of conferring protection against SARS-CoV2 morbidity and mortality, infant vaccination would result in staggered protection over time among age groups. Many countries introduced BCG vaccination between 1940 and 1950's, meaning that currently, individuals aged between 60 and 70 years would have benefitted from any potential protection it conferred to SARS-CoV2. In contrast, older age have been identified as one of the highest risk groups for being diagnosed or dying from SARS-CoV2 [30]. These data negate the causality that BCG may be conferring protection to SARS-CoV2 morbidity and mortality, at least in the older age groups.

The selection of outcome also seems to influence different conclusions on the protective effect of BCG vaccination. Some studies have suggested that SARS-CoV2 mortality alone may be influenced by BCG [31], however other studies have demonstrated that the combination of morbidity and mortality might be affected [32]. Our data showed no significant BCG effect when modelling morbidity or mortality outcomes and controlling for relevant confounding factors. Although there were differences in the degree of BCG association by outcome, this might likely be attributed to challenges in adjusting for confounding. Risk factors associated with SARS-CoV2 morbidity and transmission are challenging to control for each country as variations in environmental, individual, health system and behavioural patterns are not easy to capture. Similarly, factors associated with mortality at a country-level are complex, likely influenced by events earlier in the SARS-CoV2 care cascade. Thus, while we await evidence from ongoing randomized controlled trials (RCT) assessing various study outcomes, we cannot derive definitive conclusions on the impact of BCG on SARS-CoV2.

Our study has a number of strengths which refine current evidence around BCG vaccination and its impact on SARS-CoV2. Firstly, we conducted our analysis on a large number of countries representing a spectrum of low- to high-income countries and BCG usage. Secondly, we provided estimates of BCG association on morbidity and mortality over a two-month time horizon. This approach enabled more countries to reach significant milestones in their epidemic for comparison and allowed us to observe temporal changes in association. Thirdly, we

evaluated a number of factors implicated in SARS-CoV2 morbidity and mortality; this improved control of confounding that may have influenced previous estimates. Fourthly, we evaluated the effect of changes in restrictions following the 100[th] case and 10[th] death per country on subsequent SARS-CoV2 morbidity and mortality. Lastly, we accounted for variations in the spread of SARS-CoV2 by including a variable which measured the time taken from the 100[th] case to the analysis cut-off per country in the final model.

Our use of an ecological study design has introduced limitations to our findings. Firstly, we used BCG policy estimates in relation to infant vaccination, however there are variations to policy. In some countries, infant BCG vaccination may not be policy, however vaccination is targeted at individuals considered high-risk for TB or originating from high-burden settings; we did not however identify any literature to support this. In addition, we also noted significant variations in the duration of BCG coverage, timing of coverage, strain-type used, all of which make comparisons difficult. Secondly, our estimates of morbidity and mortality, although based on country-level information, is likely under-estimated. Evidence has shown that individuals are transmitting virus in the absence of known SARS-CoV2 symptoms, the result of which has made quantifying morbidity challenging for many countries, particularly those with limited resources, however we have attempted to adjust for possible confounding by including the number of tests per capita for each country. Cause of death may be misclassified particularly since SARS-CoV2 mortality is exacerbated among groups with comorbidities such as cardiovascular or other respiratory illnesses. Finally, the use of an ecological design introduced confounding that may not be have been controlled for. This is our biggest and most important study limitation, equally shared by some unpublished studies who have attempted to answer this question. To address this, we did include a variety of factors that might confound the protective effect of BCG vaccination; this was based on ongoing SARS-CoV2 literature and those identified as shortcomings in unpublished studies on this topic so far.

## Conclusion

The current SARS-CoV2 pandemic has challenged public health response globally. The unknown nature of the disease coupled with high morbidity and mortality rates has forced us to consider existing tools while we develop effective treatment and prevention strategies. Repurposing tools are not uncommon as they provide shorter time to implementation, however their use must be informed by evidence. In contrast to some unpublished studies, we found no association between BCG vaccination policy on SARS-CoV2 morbidity and mortality at country-level and assert that any observed affect may likely be due to uncontrolled confounding. It may be that BCG vaccination at birth does not confer immunity in later life, where it anecdotally is more likely to impact SARS-CoV2 morbidity and mortality rates, but that does not exclude the possibility that BCG vaccination later in life may be effective. Due to ecological fallacy, we cannot conclude that BCG vaccination is not associated with COVID-19 morbidity and mortality at an individual level but our findings at a national level did not find an association. While we anticipate evidence from ongoing BCG vaccination randomized controlled trials (RCTs), we urge countries to rather consider alternative tools with evidence supporting their effectiveness for controlling SARS-CoV2 morbidity and mortality such as ongoing randomized controlled trials on prevention of COVID-19 conducted in USA, Netherlands and Australia on more than 6,000 healthcare workers. The renewed prominence of BCG vaccination should also serve as a reminder for the global community that while our focus may be captured by SARS-CoV2, tuberculosis remains a significant problem, deserving greater attention and should not be neglected during this period.

## Supporting information

**S1 Table. Table on morbidity and mortality from SARS-CoV2 pandemic and the population, economic, and health characteristics of selected countries by BCG policy.**
(DOCX)

**S2 Table. Table on morbidity and mortality from SARS-CoV2 pandemic and the population, economic, and health characteristics of selected countries by BCG policy.**
(DOCX)

**S3 Table. Table on morbidity and mortality from SARS-CoV2 pandemic and the population, economic, and health characteristics of selected countries by BCG policy.**
(DOCX)

**S1 Dataset. Dataset of variables extracted from various publicly available sources.**
(XLSX)

## Acknowledgments

We would like to acknowledge and thank Helen Johns for collection of individual country data.

## Author Contributions

**Conceptualization:** Kavindhran Velen, Salome Charalambous.

**Data curation:** Salome Charalambous.

**Formal analysis:** Lucy Chimoyi.

**Methodology:** Lucy Chimoyi, James J. Lewis.

**Supervision:** James J. Lewis.

**Visualization:** Lucy Chimoyi.

**Writing – original draft:** Lucy Chimoyi, Salome Charalambous.

**Writing – review & editing:** Lucy Chimoyi, Kavindhran Velen, Gavin J. Churchyard, Robert Wallis, James J. Lewis, Salome Charalambous.

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
