## [Decision Letter · Decision Letter 0]

6 Oct 2020

PONE-D-20-26939

An ecological study to evaluate the association of Bacillus Calmette-Guerin (BCG) vaccination on cases of SARS-CoV2 infection and mortality from COVID-19

PLOS ONE

Dear Dr. Chimoyi,

Thank you for submitting your manuscript to PLOS ONE. After careful consideration, we feel that it has merit but does not fully meet PLOS ONE’s publication criteria as it currently stands. Therefore, we invite you to submit a revised version of the manuscript that addresses the points raised during the review process.

We look forward to receiving your revised manuscript.

Kind regards,

Angelo A. Izzo

Academic Editor

PLOS ONE

Journal Requirements:

3. We note that Figure 2 in your submission contain map images which may be copyrighted. All PLOS content is published under the Creative Commons Attribution License (CC BY 4.0), which means that the manuscript, images, and Supporting Information files will be freely available online, and any third party is permitted to access, download, copy, distribute, and use these materials in any way, even commercially, with proper attribution. For these reasons, we cannot publish previously copyrighted maps or satellite images created using proprietary data, such as Google software (Google Maps, Street View, and Earth). For more information, see our copyright guidelines: http://journals.plos.org/plosone/s/licenses-and-copyright.

3.1.    You may seek permission from the original copyright holder of Figure 2 to publish the content specifically under the CC BY 4.0 license. 

3.2.    If you are unable to obtain permission from the original copyright holder to publish these figures under the CC BY 4.0 license or if the copyright holder’s requirements are incompatible with the CC BY 4.0 license, please either i) remove the figure or ii) supply a replacement figure that complies with the CC BY 4.0 license. Please check copyright information on all replacement figures and update the figure caption with source information. If applicable, please specify in the figure caption text when a figure is similar but not identical to the original image and is therefore for illustrative purposes only.

Reviewers' comments:

Reviewer's Responses to Questions

**Comments to the Author**

1. Is the manuscript technically sound, and do the data support the conclusions?

Reviewer #1: Partly

Reviewer #2: Yes

2. Has the statistical analysis been performed appropriately and rigorously? 

Reviewer #1: I Don't Know

Reviewer #2: Yes

3. Have the authors made all data underlying the findings in their manuscript fully available?

Reviewer #1: Yes

Reviewer #2: Yes

4. Is the manuscript presented in an intelligible fashion and written in standard English?

Reviewer #1: Yes

Reviewer #2: Yes

5. Review Comments to the Author

Reviewer #1: Its a good descriptive analysis of prevailing COVID 19 pandemic and ecological paramenters in the sampled

countries and its association with BCG vaccination policy. However, authors are encouraged to discuss in detail

the observed association with confounding variables. Authors should clearly describe fallacy of ecological study in

ascertaining association between two variables and is descriptive in nature rather than analytical. It is appreciated

however that authors have observed caution with conclusion since confounders can mislead with both association

and non association.

1. Authors should describe in details the method of selection of countries with a flow chart.

2. They should describe the possible alternative tools being referred in conclusion.

3. Elaborate on methods used for data collection eg. immunization schedule was it standardized across the countries.

4. Provide reference for no BCG policy criteria for being used less than 30 yrs and data on current rates of BCG immunization

in BCG policy states. classification of current, ever and never

5. Please explain why were the countries with less than 1 million population excluded.

6. Why weren't control countries (with limited or no COVID cases) considered in the study for comparison.

7. Significant association of COVID cases with population and testing rates, tabulated with BCG policy fo some understanding.

8. Figure 2 is not well explanatory, Kindly redesign.

9. Unpublished studies mentioned should be referred or identified.

Reviewer #2: These are my comments about the manuscript "An ecological study to evaluate the association of Bacillus Calmette-Guerin (BCG) vaccination on cases of SARS-CoV2 infection and mortality from COVID-19" that I was asked to evaluate.

The authors report the results of an ecological study aimed at evaluating the association between national BCG vaccination status and rates of COVID-19 incidence and mortality. They do so by utilizing national-level data from various sources while, implementing log-linear regression models, adjusting for a host of factors that could be confounders or mediators. They conclude that there is no association between BCG status and COVID-19.

First, the topic is of great importance on a global level, and much has been written about the theoretical protective effect of BCG against severe COVID-19. These speculations have even initiated several large clinical studies, as well as mass vaccination in Japan that has caused supply shortages of BCG. For that reason, I believe that, though the topic is not novel, it is still of great interest to the scientific community.

The manuscript is well written, and the internal flow is clear. The use of ecological studies has been greatly criticized and is inherently open to confounding. The authors have gone to great lengths to minimize this effect by adjustments to a host of variables and by selecting outcomes that are less prone to bias, but it is still an ecological study.

My main comment would be that it seems that some time has passed since the manuscript was finalized, and several papers have been published on the same topic, none of which is mentioned and discussed, leaving the manuscript somewhat outdated. Just to give a few examples, of many others:

Lindestam Arlehamn CS, Sette A, Peters B. Lack of evidence for BCG vaccine protection from severe COVID-19. Proc Natl Acad Sci U S A. 2020 Sep 29:202016733. doi: 10.1073/pnas.2016733117. Online ahead of print. PMID: 32994350

de Chaisemartin C, de Chaisemartin L BCG vaccination in infancy does not protect against COVID-19. Evidence from a natural experiment in Sweden. Clin Infect Dis. 2020 Aug 23:ciaa1223. doi: 10.1093/cid/ciaa1223. Online ahead of print. PMID: 32829400

Hamiel U, Kozer E, Youngster I. SARS-CoV-2 Rates in BCG-Vaccinated and Unvaccinated Young Adults. JAMA. 2020 Jun 9;323(22):2340-2341. doi: 10.1001/jama.2020.8189. PMID: 32401274

Thus, I would recommend extensively revising and updating the background and discussion to reflect this, both in terms of content and in terms of references.

Thank you for allowing me to review your work

6. PLOS authors have the option to publish the peer review history of their article (what does this mean?). If published, this will include your full peer review and any attached files.

Reviewer #1: No

Reviewer #2: No

---

## [Author Response · Author response to Decision Letter 0]

6 Nov 2020

PLEASE FIND MY RESPONSES IN CAPS

THE MANUSCRIPT MAIN BODY HAS BEEN REVISED ACCORDING TO THE JOURNAL REQUIREMENTS

THE TITLE PAGE HAS BEEN REVISED ACCORDING TO THE SPECIFIED REQUIREMENTS

WE HAVE REVIEWED THE MANUSCRIPT AND HAVE EDITED THE CONTENTS FOR LANGUAGE USE, GRAMMAR AND SPELLING.

3. We note that Figure 2 in your submission contain map images which may be copyrighted. All PLOS content is published under the Creative Commons Attribution License (CC BY 4.0), which means that the manuscript, images, and Supporting Information files will be freely available online, and any third party is permitted to access, download, copy, distribute, and use these materials in any way, even commercially, with proper attribution. For these reasons, we cannot publish previously copyrighted maps or satellite images created using proprietary data, such as Google software (Google Maps, Street View, and Earth). For more information, see our copyright guidelines: http://journals.plos.org/plosone/s/licenses-and-copyright.

THIS MAP HAS NOT BEEN COPYRIGHTED FROM ANY SOURCE. IT HAS BEEN CREATED WITH PUBLICLY AVAILABLE INFORMATION BY THE AUTHOR (LC)

 N/A

3.1. You may seek permission from the original copyright holder of Figure 2 to publish the content specifically under the CC BY 4.0 license. 

 N/A

 N/A

N/A

3.2. If you are unable to obtain permission from the original copyright holder to publish these figures under the CC BY 4.0 license or if the copyright holder’s requirements are incompatible with the CC BY 4.0 license, please either i) remove the figure or ii) supply a replacement figure that complies with the CC BY 4.0 license. Please check copyright information on all replacement figures and update the figure caption with source information. If applicable, please specify in the figure caption text when a figure is similar but not identical to the original image and is therefore for illustrative purposes only.

N/A

THIS HAS BEEN INCLUDED IN THE MANUSCRIPT BEFORE THE REFERENCES. WE HAVE USED PACE TO ENSURE THE FIGURES MEET THE JOURNAL REQUIREMENTS

The responses to the reviewers comments are as follows;

It is a good descriptive analysis of prevailing COVID 19 pandemic and ecological parameters in the sampled countries and its association with BCG vaccination policy. However, authors are encouraged to discuss in detail the observed association with confounding variables. 

THANK YOU FOR THIS OBSERVATION. WE HAVE INTERPRETED THE SIGNIFICANT ASSOCIATIONS OF COVID-19 CASES WITH OTHER CO-VARIATES IN THE DISCUSSION SECTION IN LINES 238-245 ON PAGE 13.

Authors should clearly describe fallacy of ecological study in ascertaining association between two variables and is descriptive in nature rather than analytical. It is appreciated however that authors have observed caution with conclusion since confounders can mislead with both association and non-association.

WE HAVE DESCRIBED THE CONCEPT OF ECOLOGICAL FALLACY IN THE DISCUSSION SECTION OF THE REVISED MANUSCRIPT IN LINES 338 AND 340 ON PAGE 16.

Authors should describe in details the method of selection of countries with a flow chart.

A FLOW CHART (FIG 1) DESCRIBING COUNTRY SELECTION HAS BEEN INCLUDED.

They should describe the possible alternative tools being referred in conclusion.

THE ALTERNATIVE TOOLS ARE RANDOMIZED CONTROLLED TRIALS WHICH IS ADDED IN LINES 332 AND 333 ON PAGE 16.

Elaborate on methods used for data collection e.g. immunization schedule was it standardized across the countries.

WE USED STANDARDISED DATABASES FOR MOST INDICATORS. WE HAVE INCLUDED THE WORD STANDARDIZED IN LINE 134 OF PAGE 5. FOR ALL INDICATORS THE SOURCES ARE SHOWN IN TABLE 1. 

Provide reference for no BCG policy criteria for being used less than 30 years and data on current rates of BCG immunization in BCG policy states. Classification of current, ever and never

THE REFERENCE FOR BCG CRITERIA HAS BEEN INCLUDED ON PAGE 5; LINE 150.

Please explain why the countries with less than 1 million population excluded.

THIS WAS BECAUSE WE FELT THAT TRANSMISSION IN A VERY SMALL COUNTRY MAY BE AFFECTED IN A DIFFERENT WAY TO LARGER POPULATIONS. THIS HAS BEEN ADDED AT THE END OF THE PARAGRAPH ON PAGE 5; LINES 119 AND 120.

IN MOST CASES THOSE COUNTRIES ALSO DID NOT HAVE INFORMATION ABOUT A BCG POLICY AND SO WOULD HAVE BEEN EXCLUDED DUE TO OTHER CRITERIA AS WELL.

Why weren't control countries (with limited or no COVID cases) considered in the study for comparison?

THE ECOLOGICAL ANALYSIS INCLUDED COUNTRIES WITH NO/LOW COVID-19 AGAINST THOSE WITH HIGH CASES/DEATHS. WE DID NOT SET OUT TO DO A COMPARISON.

Significant association of COVID cases with population and testing rates, tabulated with BCG policy for some understanding.

THANK YOU FOR THIS OBSERVATION. WE HAVE INTERPRETED THE SIGNIFICANT ASSOCIATIONS OF COVID-19 CASES WITH OTHER CO-VARIATES IN THE RESULTS AND EXPLAINED THIS ASSOCIATION IN THE DISCUSSION SECTION. 

Figure 2 is not well explanatory, Kindly redesign.

THE MAP HAS BEEN REDESIGNED. WE HAVE CHANGED SHOWN THE VARIED DISTRIBUTION OF STRINGENCY MEASURES AT COUNTRY LEVEL USING SOLID COLOURS AS OPPOSED TO HASH DESIGNS. RED INDICATING LOW LEVELS (<25%) MEASURES AND GREEN SHOWING HIGH LEVELS (>75%) OF STRINGENCY MEASURES IMPLEMENTED BY THE 100TH REPORTED CASE.

Unpublished studies mentioned should be referred or identified.

THANK YOU FOR THIS OBSERVATION. THE REFERENCES HAVE BEEN ADDED AT THE END OF THIS STATEMENT AND THE REVISION IS FOUND ON PAGE 4; LINES 115 AND 116.

Reviewer 2

These are my comments about the manuscript "An ecological study to evaluate the association of Bacillus Calmette-Guerin (BCG) vaccination on cases of SARS-CoV2 infection and mortality from COVID-19" that I was asked to evaluate.

The authors report the results of an ecological study aimed at evaluating the association between national BCG vaccination status and rates of COVID-19 incidence and mortality. They do so by utilizing national-level data from various sources while, implementing log-linear regression models, adjusting for a host of factors that could be confounders or mediators. They conclude that there is no association between BCG status and COVID-19. First, the topic is of great importance on a global level, and much has been written about the theoretical protective effect of BCG against severe COVID-19. These speculations have even initiated several large clinical studies, as well as mass vaccination in Japan that has caused supply shortages of BCG. For that reason, I believe that, though the topic is not novel, it is still of great interest to the scientific community.

EVEN THOUGH THE TOPIC IS NOT NOVEL, WE PROVED THAT THE APPARENT ASSOCIATION REPORTED EARLY ON IN THE EPIDEMIC HAD NOT TAKEN INTO ACCOUNT OTHER COUNTRY-LEVEL SPECIFIC FACTORS. WE FURTHER INVESTIGATED THIS ASSOCIATION OVER FOUR TIME PERIODS AND OUR FINDINGS SUGGESTED THAT AS THE TIME PERIOD PROGRESSED, THE ASSOCIATION WEAKENED AND WAS INSIGNIFICANT. THIS HAS BEEN EMPHASISED AS A STRENGTH OF THE PAPER.

The manuscript is well written, and the internal flow is clear. The use of ecological studies has been greatly criticized and is inherently open to confounding. The authors have gone to great lengths to minimize this effect by adjustments to a host of variables and by selecting outcomes that are less prone to bias, but it is still an ecological study. My main comment would be that it seems that some time has passed since the manuscript was finalized, and several papers have been published on the same topic, none of which is mentioned and discussed, leaving the manuscript somewhat outdated. 

Just to give a few examples, of many others:

Lindestam Arlehamn CS, Sette A, Peters B. Lack of evidence for BCG vaccine protection from severe COVID-19. Proc Natl Acad Sci U S A. 2020 Sep 29:202016733. doi: 10.1073/pnas.2016733117. Online ahead of print. PMID: 32994350

de Chaisemartin C, de Chaisemartin L BCG vaccination in infancy does not protect against COVID-19. Evidence from a natural experiment in Sweden. Clin Infect Dis. 2020 Aug 23:ciaa1223. doi: 10.1093/cid/ciaa1223. Online ahead of print. PMID: 32829400

Hamiel U, Kozer E, Youngster I. SARS-CoV-2 Rates in BCG-Vaccinated and Unvaccinated Young Adults. JAMA. 2020 Jun 9;323(22):2340-2341. doi: 10.1001/jama.2020.8189. PMID: 32401274

Thus, I would recommend extensively revising and updating the background and discussion to reflect this, both in terms of content and in terms of references.

WE APPRECIATE THE POSITIVE COMMENTS FROM THE REVIEWER ON COHERENCE OF THE STUDY. WE THANK YOU FOR EXAMPLES OF PUBLISHED WORK WHICH WE HAVE INCLUDED THESE TOGETHER WITH OTHER ARTICLES TO UPDATE OUR DISCUSSION SECTION LINES 248-256; PAGE 13.

---

## [Decision Letter · Decision Letter 1]

30 Nov 2020

An ecological study to evaluate the association of Bacillus Calmette-Guerin (BCG) vaccination on cases of SARS-CoV2 infection and mortality from COVID-19

PONE-D-20-26939R1

Dear Dr. Chimoyi,

We’re pleased to inform you that your manuscript has been judged scientifically suitable for publication and will be formally accepted for publication once it meets all outstanding technical requirements.

Kind regards,

Angelo A. Izzo

Academic Editor

PLOS ONE

Additional Editor Comments (optional):

Reviewers' comments:

Reviewer's Responses to Questions

**Comments to the Author**

1. If the authors have adequately addressed your comments raised in a previous round of review and you feel that this manuscript is now acceptable for publication, you may indicate that here to bypass the “Comments to the Author” section, enter your conflict of interest statement in the “Confidential to Editor” section, and submit your "Accept" recommendation.

Reviewer #1: (No Response)

Reviewer #2: All comments have been addressed

2. Is the manuscript technically sound, and do the data support the conclusions?

Reviewer #1: (No Response)

Reviewer #2: Yes

3. Has the statistical analysis been performed appropriately and rigorously? 

Reviewer #1: (No Response)

Reviewer #2: Yes

4. Have the authors made all data underlying the findings in their manuscript fully available?

Reviewer #1: (No Response)

Reviewer #2: Yes

5. Is the manuscript presented in an intelligible fashion and written in standard English?

Reviewer #1: (No Response)

Reviewer #2: Yes

6. Review Comments to the Author

Reviewer #1: (No Response)

Reviewer #2: The authors have addressed my concerns; I have no further comments.

Thank you for allowing me to review your work

7. PLOS authors have the option to publish the peer review history of their article (what does this mean?). If published, this will include your full peer review and any attached files.

Reviewer #1: No

Reviewer #2: No

---

## [Editor Report · Acceptance letter]

9 Dec 2020

PONE-D-20-26939R1 

An ecological study to evaluate the association of Bacillus Calmette-Guerin (BCG) vaccination on cases of SARS-CoV2 infection and mortality from COVID-19 

Dear Dr. Chimoyi:

I'm pleased to inform you that your manuscript has been deemed suitable for publication in PLOS ONE. Congratulations! Your manuscript is now with our production department. 

Kind regards, 

on behalf of

Dr. Angelo A. Izzo 

Academic Editor

PLOS ONE